# Development and testing of a composite index to monitor the continuum of maternal health service delivery at provincial and district level in South Africa

**Mamothena Carol Mothupi**[1]*, **Jeroen De Man**[2], **Hanani Tabana**[1], **Lucia Knight**[3]

**1** School of Public Health, Faculty of Community and Health Sciences, University of the Western Cape, Cape Town, South Africa, **2** Department of Primary and Interdisciplinary Care, Centre for General Practice, University of Antwerp, Antwerp, Belgium, **3** School of Public Health & Family Medicine, Faculty of Health Sciences, University of Cape Town, Cape Town, South Africa

* mamothena@gmail.com

## Abstract

### Introduction

The continuum of care is a recommended framework for comprehensive health service delivery for maternal health, and it integrates health system and social determinants of health. There is a current lack of knowledge on a measurement approach to monitor performance on the framework. In this study we aim to develop and test a composite index for assessing the maternal health continuum in a province in South Africa with the possibility of nationwide use.

### Materials and methods

The composite index was computed as a geometric mean of four dimensions of adequacy of the continuum of care. Data was sourced from the district health information system, household surveys and the census. The index formula was tested for robustness when alternative inputs for indicators and standardization methods were used. The index was used to assess performance in service delivery in the North West province of South Africa, as well as its four districts over a five-year period (2013–2017). The index was validated by assessing associations with maternal health and other outcomes. And factor analysis was used to assess the statistical dimensions of the index.

### Results

The provincial level index score increased from 62.3 in 2013 to 74 in 2017, showing general improvement in service delivery over time. The district level scores also improved over time, and our analysis identified areas for performance improvement. These include social determinants of health in some districts, and access and linkages to care in others. The provincial index was correlated with institutional maternal mortality rates ($r_s$ = -0.90, 90% CI = (-1.00, -0.25)) and the Human Development Index (r = 0.97, 95% CI = (0.63, 0.99). It was robust to

**Data Availability Statement:** District Health Information System data cannot be shared publicly because of Department of Health restrictions. Data are available from the North West Department of

Health South Africa Institutional Data Access / Ethics Committee (contact via Dr. Frikkie Reichel FReichel@nwpg.gov.za) for researchers who meet the criteria for access to confidential data. The data underlying the results presented in the study are available from (DataFirst – University of Cape Town and https://www.datafirst.uct.ac.za/dataportal/index.php/catalog).

**Funding:** MM received funding to conduct this study. This work is based on the research supported by the South African Research Chairs Initiative of the Department of Science and Technology (https://www.dst.gov.za/) and National Research Foundation (https://www.nrf.ac.za/) of South Africa (Grant No. 82769). MM would also like to acknowledge funding from the South African Medical Research Council(https://www.samrc.ac.za/) and the Belgian Development Cooperation, through the Institute of Tropical Medicine(https://www.itg.be/) Antwerp. Any opinion, finding and conclusion or recommendation expressed in this material is that of the authors and not the funders. The funders did not have a role in the design of the study and collection, analysis, and interpretation of data and in writing the manuscript.

**Competing interests:** The authors have declared that no competing interests exist.

alternative approaches including z-score standardization of indicators. Factor analysis showed three groupings of indicators for the health system and social determinants of health.

## Conclusions

This study demonstrated the development and testing of a composite index to monitor and assess service delivery on the continuum of care for maternal health. The index was shown to be robust and valid, and identified potential areas for service improvement. A contextualised version can be tested in other settings within and outside of South Africa.

## Introduction

Maternal health outcomes in South Africa (SA) remain poor despite national investments toward their improvement [1, 2]. The maternal mortality ratio (MMR) was estimated between 138 and 158 deaths per 100,000 live births in 2015 [3]. Pregnancy related and facility-based rates of maternal mortality are also high, estimated in 2016 at 536 per 100,000 and 135 per 100,000 respectively [2, 4]. Maternal mortality refers to deaths occurring during pregnancy up to 42 days post-partum or termination of pregnancy, from a pregnancy-related cause; and pregnancy related deaths include all deaths in that period, where cause of death is not identified [5, 6]. The major causes of maternal mortality include HIV infection, obstetric haemorrhage, and hypertensive disorders [7]. The prevailing challenges in maternal health in SA, which also contribute to death, include: inequalities in health service access [8], poor coverage and quality of essential interventions [9], inadequate system wide improvements in quality of care [2] and weak community health services [2]. As such, health systems, health worker and patient related factor are important determinants of individual morbidity and mortality.

One of the key strategies to address maternal health challenges in SA has been to strengthen service provision at all levels of care, from the community to the regional and tertiary hospitals [10–12]. The continuum of care for maternal and child health is a public health framework for outlining the essential interventions and addressing service delivery challenges [13, 14]. The framework has been developed for low- and middle income countries (LMICs) [13] and adapted by national health system stakeholders to the South African context, as illustrated in Fig 1.

The framework outlines interventions from pre-pregnancy to childhood; the maternal health interventions encompass reproductive health, antenatal, delivery and postnatal care. The framework for SA also outlines "intersectoral factors", which represent social determinants of health such as housing, nutrition, water and sanitation, and education (Fig 1). The implementation of the framework is expected to improve health outcomes by improving coverage and comprehensiveness of services, quality of clinical care, mitigating duplication of resources, and improving integration of health services [13, 15, 16].

A crucial barrier to the implementation of the continuum of care framework in SA and many LMICs is the lack of a comprehensive monitoring tool for service delivery [11, 17]. The current discourse in SA focuses on the importance of integrated delivery of services, quality of care, strengthening community health systems, and multisectoral collaboration to improve outcomes [1, 11, 15, 18, 19]. Maternal death audits have emphasized the importance of referral linkages, women's empowerment, quality of care (including quality antenatal care, prevention of pregnancy associated hypertension etc.), and post-natal follow-ups to improve maternal health outcomes in the country [20]. However, gaps remain in measuring community and social factors influencing maternal health outcomes [21]. Assessment of the continuum of care

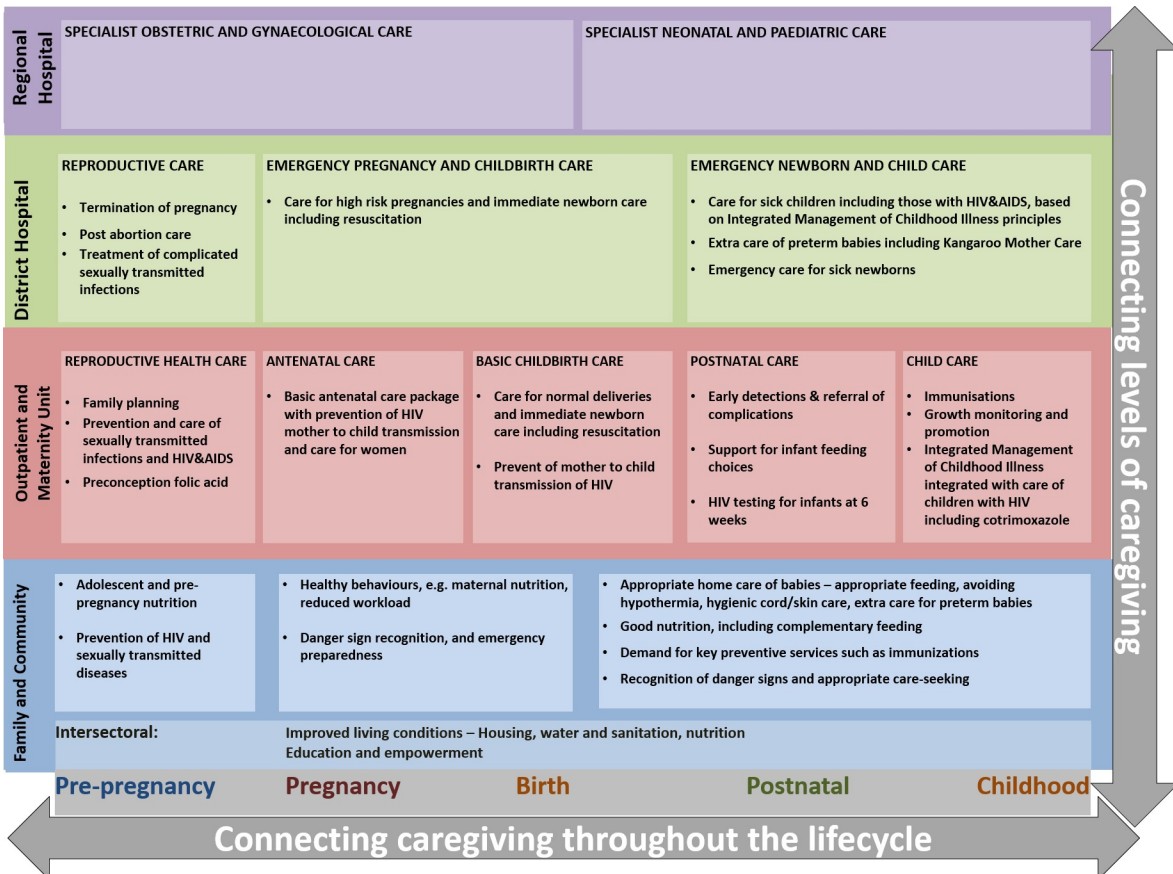

**Fig 1. The continuum of care framework for maternal, new-born and child health in South Africa [10].**

requires consideration of a broad set of indicators beyond antenatal, birth and postnatal care [17].

Previous studies by these authors reviewed and evaluated available indicators for tracking services on the continuum of care for maternal health in South Africa [22, 23]. Another study by these authors proposed an analytical approach emphasizing assessment of access and utilization, quality of care, linkages of care and social determinants of health [17]. Multidimensional assessment is often carried out with composite indices that summarize the performance of multiple interventions on the continuum. Composite indices have been used to track continuum of care performance at subnational and global levels [24–26], while a gap remains in broader integration of quality and social determinants of health [17, 27]. In this study we explore an approach to combine a broad set of indicators for maternal health interventions on the continuum of care through development of a composite index. We explore if the index can be used to assess service delivery at subnational levels in SA and the implications for future monitoring efforts to support implementation of the framework.

## Methods

### Setting

The North West province is one of the nine SA provinces and consists of four districts: Dr Kenneth Kaunda, Ngaka Modiri Molema, Dr Ruth Segomotsi Mompati, and Bojanala

Platinum District Municipalities. This province is among the worst performers with regards to maternal health outcomes and health system indicators [28]. However, the province was also one of the pioneers of primary health care quality improvement and health system strengthening initiatives, such as the Ward Based Outreach and the primary level Ideal Clinic realization programs [29, 30]. Thus, the province was expected to have a broader set of available indicators across the continuum of care compared to others.

## Design

We used a step-wise approach to develop and test the proposed index, based on current methodological guidelines [31, 32]. The main steps include: i) defining a theoretical/conceptual framework, ii) selection of variables/indicators, iii) imputation of missing data, iv) multivariate analysis, v) scaling of indicators, vi) weighting and aggregation, vii) checking for robustness, and viii) validation [31].

## Conceptual framework

A critical interpretive synthesis of current measurement and monitoring approaches in LMICs found a gap in multi-dimensionality of sets of indicators currently used to assess the continuum of care (COC) for maternal health [17]. The *adequacy construct* was therefore defined, which outlines four important dimensions to consider: 1) access and utilization of care; 2) quality of care; 3) linkages between levels and packages of care; and 4) social determinants of health [17] (Fig 2). The adequacy construct complements the COC framework by adding elements of quality of and linkages to care, and proposing that all four dimensions be monitored, not just access and or utilization. Indicators of service delivery across all dimensions should therefore be sought from local data sources, with consideration for their relevance, feasibility, validity and quality of routinely collected data such as the District Health Information System [22, 23].

## Selection of variables/indicators

Our previous research has described the systematic process for selection of indicators and assessment of suitability and measurement gaps [22, 23]. The indicators were aligned with the interventions along the continuum of care framework as illustrated in Fig 1, thus reflecting mainly inputs, processes and outputs in service delivery (outcomes indicators were used during validation steps of the index development process). Selection was also based on their

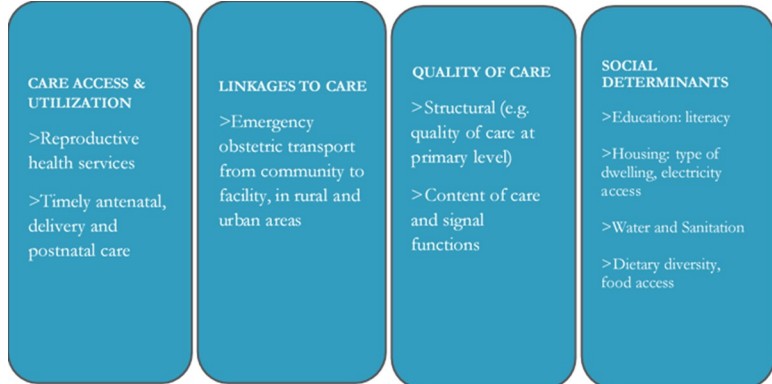

**Fig 2. Dimensions and types of indicators used to develop the continuum of care index for maternal health.**

availability within the South African data sources. These indicators were extracted for the North West province and districts for the period 2013–2017. Health service indicators were sourced from the National Indicator Data Set (NIDS) of the District Health Information System (DHIS). The DHIS is used to report and monitor facility level data for health services to support policy and planning [33]. The DHIS provided indicators for access and utilization, linkages, and quality of care dimensions of the continuum of care. This was done with consideration for the limitations of DHIS data due to poor quality of routinely collected data in South Africa and other LMICs contexts.

Social determinant indicators were sourced from the annual *General Household Survey* [34] *(GHS) (2013–2017)* [34], *Census 2011* [35] and *Community Survey (CS) 2016* [36]. The census and CS enabled assessment at district level, even though they offer fewer social determinants of health indicators than the GHS. The census and CS provided indicators of literacy, housing, access to electricity, water, and sanitation at the district level. Additionally, the CS also assesses dietary behaviour and empowerment, but this was not included in the final analysis of performance to allow district level comparison with the census indicator set. There were no comparable indicators of dietary behaviour and empowerment across the data sources at the district level, so only those applicable across were included. A description of all indicators used in this study is provided in S1 Table. Indicator data were extracted, cleaned, and analysed in MS Excel 2010, R v3.6.1 and STATA 14.0.

## Imputation of missing data

Health service indicator data may be missing due to lack of services and under-performing systems for data collection and reporting which affects data quality. In the period between 2013–2017 reporting rates gradually improved from a median under 20% to over 75% across most districts. Missing data was also common in the beginning of the period and improved gradually over time. These are systematic issues that are considered to affect availability of data for indicators completely at random. While there is a need to strengthen the health information system to improve data quality, the authors did some corrections for missing data in order to conduct this analysis. We conducted single value imputation using the indicator value observed from the adjacent year [37]. In the Results section we discuss the impact of the remaining data gaps on the index findings. Single value imputation was also applied to indicators from community survey and census to allow calculation of index at district level.

## Multivariate analysis

We used exploratory factor analysis to assess dimensionality of the data, in order to compare the statistical and conceptual groupings of indicators [31]. We assessed whether the data fitted the four dimensions of continuum of care proposed by the adequacy construct. The output of the exploratory factor analysis is assessed in the Results section.

## Scaling of indicators

We developed a scale transformation approach of scoring indicator values on a scale between 0 and 100 [31, 38, 39] (Eq 1).

$$\text{Indicator score} = \text{Ideal Score} - |(\text{Target} - \text{Performance}) * 100| \qquad (1)$$

The **indicator score** is calculated on a scale between 0–100; the **ideal score** is the maximum attainable score, which is a 100; the **target** is the ideal performance of the indicator; and **performance** is the observed value of the indicator during a given time period. Targets may consist of a range of values and in such a case we calculated the median score to represent

indicator performance. Targets were also based on national policy documents and global technical or scientific guidelines [31, 32, 38]. Targets were set to the conservative maximum (100%) where guidelines were unavailable. The difference between target and observed performance is multiplied by 100 because indicators are originally measured as percentages/proportions. Using targets for performance improves the meaningfulness of the index and its' role in policy discourse [39].

## Weighting and aggregation

The comprehensive continuum of care for maternal health index ($C_3MH$ index) was computed as a geometric mean of equally weighted sub-indices reflecting the four adequacy dimensions. We chose equal weighting since this was estimated the most reasonable approach, and based on lack of evidence on the relative importance of each sub-index, lack of theoretical structure to justify a differential weighting scheme, and inadequate statistical and/or empirical knowledge, among others [31, 32, 40, 41].

$$C_3MH \text{ index} = (\text{Access to care} \cdot \text{Linkages} \cdot \text{Quality of Care Social Determinants of Health})^{1/4} (2)$$

Simple indices, based on arithmetic and geometric means, can be robust and give valuable information about public health or health system performance [25, 26, 40, 42, 43]. Unlike the arithmetic mean, the geometric mean allows for a degree of non-compensation of performance of one indicator by another [31, 32]. Each sub-index (e.g. access to care) was also formulated as a geometric mean of its indicator scores.

$$\text{Sub index score} = (\text{Indicator}_a \cdot \text{Indicator}_b \cdot \text{Indicator}_c \cdots \text{Indicator}_n)^{1/n} \qquad (3)$$

Where $a$, $b$, $c$ are individual indicators and $n$ = number of indicators within the sub-index.

## Validity and robustness

We ran sensitivity analyses comparing index performance with different indicator combinations and normalization methods [44]. We tested if z-score standardization leads to a shift in district ranks [31]. Index performance was also compared after removal of indicators that were considered outliers (performed close to 100%), missing data, or indicators that could be represented by a proxy (e.g. syphilis treatment measured with one indicator instead of the three across the treatment cascade, see S1 Table Indicators 4–6). Index aggregation by arithmetic and geometric mean was also compared. We assessed the median absolute difference in district ranks, and its inter-quartile range, testing alternative approaches to index formulation [44]. External validation of the index was conducted by exploring its relationship with indicators of public health performance and maternal health outcomes, particularly the Human Development Index (HDI) and maternal mortality rates [38, 45, 46]. Confidence intervals for correlations were calculated by bootstrapping methods in R v3.6.1.

## Results

### Performance at provincial level

In the North West province, we combined 12 indicators of access and utilization to care, two quality of care, two linkages of care, and 9 social determinants of health indicators to measure the $C_3MH$ index (Table 1).

The $C_3MH$ index at the provincial level ranged from 62.3 in 2013 to 74 in 2017, showing a general trend of improvement (see Table 1). The two sub-indices that substantially contributed to this increase were:

**Table 1. The continuum of care for maternal health index, sub- indices and indicators for North West Province, South Africa, in the period 2013–2017.**

| Indicators | Targets | Indicator Performance | | | | | Indicator Scores | | | | |
|---|---|---|---|---|---|---|---|---|---|---|---|
| | | 2013 | 2014 | 2015 | 2016 | 2017 | 2013 | 2014 | 2015 | 2016 | 2017 |
| Cervical cancer screening coverage | 100% | 58% | 62% | 66% | 66% | 69% | 58 | 62 | 66 | 66 | 69 |
| Antenatal 1st visit before 20 weeks rate | 100% | 48% | 53% | 59% | 64% | 64% | 48 | 53 | 59 | 64 | 64 |
| Antenatal 1st visit coverage | 100% | 78% | 79% | 75% | 76% | 78% | 78 | 79 | 75 | 76 | 78 |
| Syphilis positive pregnant female receive Benz-penicillin 1st dose rate | 100% | na | na | 57% | 57% | 78% | na | na | 57 | 57 | 78 |
| Syphilis positive pregnant female receive Benz-penicillin 2nd dose rate | 100% | na | na | 60% | 60% | 57% | na | na | 60 | 60 | 57 |
| Syphilis positive pregnant female receive Benz-penicillin 3rd dose rate | 100% | na | na | 57% | 57% | 50% | na | na | 57 | 57 | 50 |
| Antenatal client starts on antiretroviral therapy rate | 100% | 63% | 88% | 89% | 93% | 93% | 63 | 88 | 89 | 93 | 93 |
| Delivery by Caesarean section rate | 5–15% | 18% | 21% | 22% | 21% | 24% | 92 | 89 | 88 | 89 | 86 |
| Delivery in facility rate | 100% | 65% | 67% | 69% | 69% | 72% | 65 | 67 | 69 | 69 | 72 |
| Mother postnatal visit within 6 days rate | 80–100% | 75% | 76% | 71% | 73% | 77% | 85 | 86 | 81 | 83 | 87 |
| Couple year protection rate | 50–100% | 42% | 54% | 50% | 59% | 55% | 67 | 75 | 75 | 75 | 75 |
| Termination of pregnancy 0–12 weeks rate | 100% | 97% | 96% | 95% | 95% | 96% | 97 | 96 | 95 | 95 | 96 |
| *Access sub-index* | | | | | | | *70.9* | *76.0* | *71.5* | *72.5* | *74.1* |
| Antenatal client HIV re-test rate | 100% | 47% | 64% | 78% | 100% | 100% | *47* | *64* | *78* | *100* | *100* |
| Average ideal clinic status (score) | 70–100% | na | 55% | 55% | 65% | 66% | *na* | *70* | *70* | *80* | *81* |
| *Quality sub-index* | | | | | | | *47* | *66.9* | *73.9* | *89.4* | *90.0* |
| Emergency rural obstetric response under 40 minutes rate | 75–100% | na | na | na | 61% | 61% | *na* | *na* | *na* | *74* | *74* |
| Emergency urban obstetric response under 15 minutes rate | 75–100% | na | na | na | 41% | 41% | *na* | *na* | *na* | *54* | *54* |
| *Linkages sub-index* | | | | | | | *na* | *na* | *na* | *62.7* | *62.7* |
| Domestic water compliance rate | 100% | 72% | 53% | 62% | 76% | 63% | *72* | *53* | *62* | *76* | *63* |
| % women 15–49 who are literate | 100% | 83% | 84% | 83% | 83% | 82% | *83* | *84* | *83* | *83* | *82* |
| % women 15–49 in households with adequate water infrastructure | 100% | 82% | 82% | 78% | 80% | 80% | *82* | *82* | *78* | *80* | *80* |
| % women 15–49 with basic sanitation facility | 100% | 71% | 69% | 70% | 71% | 71% | *71* | *69* | *70* | *71* | *71* |
| % women 15–49 living in adequate housing | 100% | 52% | 44% | 48% | 54% | 55% | *52* | *44* | *48* | *54* | *55* |
| % women 15–49 living in formal housing | 100% | 85% | 85% | 82% | 82% | 83% | *85* | *84* | *82* | *82* | *83* |
| % women 15–49 with access to electricity | 100% | 95% | 95% | 94% | 94% | 95% | *95* | *95* | *94* | *94* | *95* |
| % women 15–49 who have adequate food access | 100% | 62% | 62% | 62% | 64% | 64% | *62* | *62* | *62* | *64* | *64* |
| Mean Household Dietary Diversity Score (women 15–49) (converted to 100) | 100% | 62 | 61 | 62 | 62 | 62 | *62* | *61* | *62* | *62* | *62* |
| *SDoH index score* | | | | | | | *72.6* | *68.6* | *69.9* | *73.0* | *71.8* |
| *CoC (maternal health) Index* | | | | | | | *62.3* | *70.4* | *71.7* | *73.8* | *74.0* |

SDoH = social determinants of health; CoC = continuum of care.

- *Improvement in access and utilization of care indicators*, particularly cervical cancer screening, timely antenatal care, and antiretroviral drug provision.

- *Improvement in facility performance on quality of care measures*. The quality sub-index improved from 66.9 in 2014 to 90 in 2017. However, this sub-index is only based on two indicators, one of which represents a drastic improvement in HIV program processes. A gap exists in maternal health care safety and patient experience of care indicators for measurement of quality in the North West province.

Little overall improvement was made in the social determinants of health during that period, which may point to a slow pace of development in the province. Data was unavailable for the period (2013–14) to monitor treatment of sexually transmitted illness (syphilis), emergency obstetric transport (2013–2015), and quality of care (Ideal Clinic) (2013–2014, while the program was under conceptualization and testing).

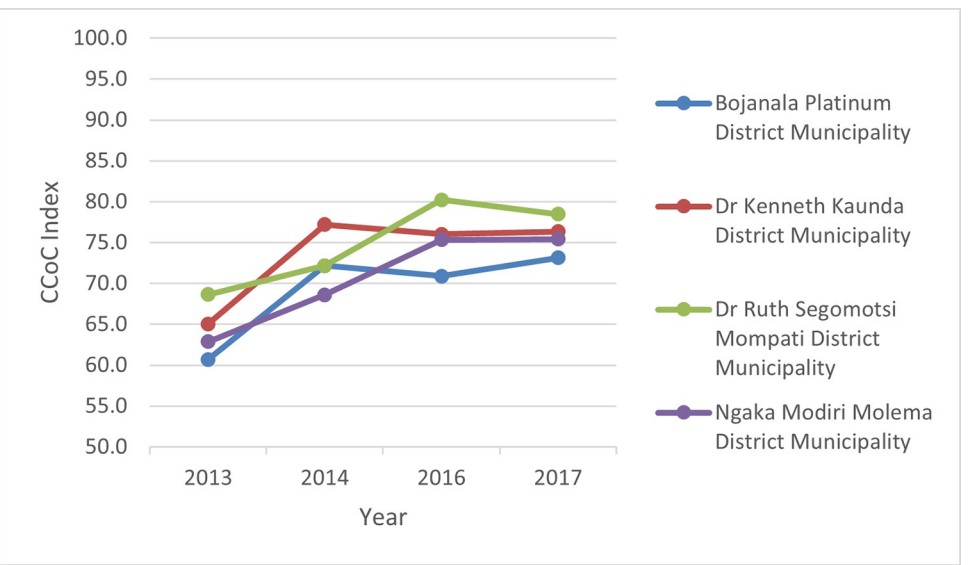

**Fig 3. Comprehensive continuum of care index (C3MHindex) scores by districts over a five-year period 2013–2017.**

## Monitoring performance at district level

There was an overall improvement in the index at district level over the period 2013–2017, as illustrated in Fig 3.

Overall, Dr Ruth Segomotsi Mompati (RSM) district performed better than other districts on the index, while Bojanala Platinum performed generally poorer. We also compared sub-index performance to demonstrate effect on overall scores, using 2016 as a reference year (Fig 4).

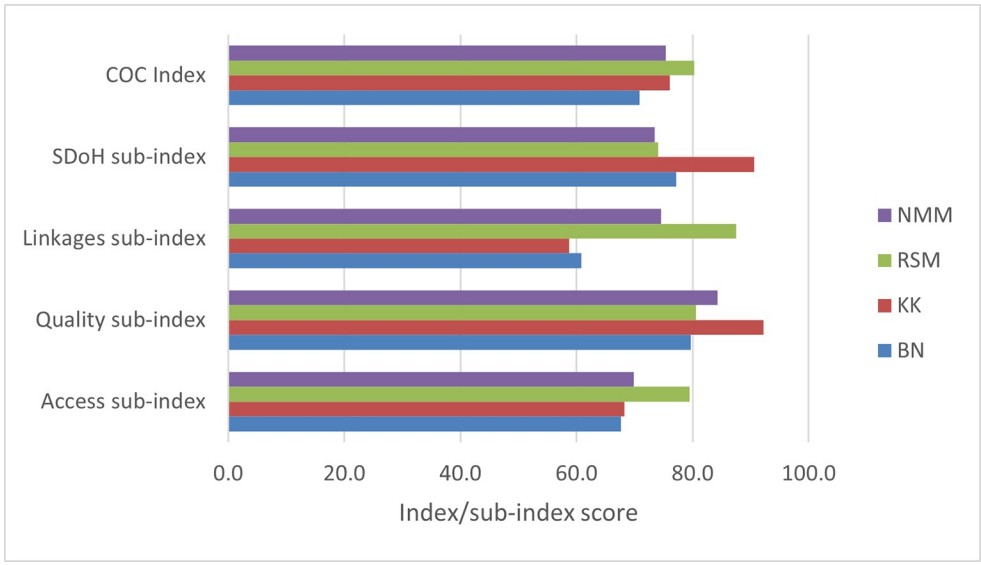

**Fig 4. Sub-index and C₃MH index scores by districts in 2016.** BN = Bojanala Platinum District, KK = Dr Kenneth Kaunda District Municipality, RSM = Dr Ruth Segomotsi Mompati District Municipality, NMM = Ngaka Modiri Molema District Municipality, SDOH = Social determinants of health.

**Table 2. Spearman rank correlation between alternatives for indicator standardization and aggregation at district level.**

|  | Base case | z-score (districts) | arithmetic mean |
|---|---|---|---|
| **Base case** | 1.00 |  |  |
| **z-score** | 0.83 | 1.00 |  |
| **arithmetic mean (d)** | 0.95 | 0.84 | 1.00 |

Base case is based on linear scaling (our method) and geometric mean aggregation.

In 2016, Dr Kenneth Kaunda district scored relatively higher than other districts on the social determinants of health and quality of care sub-indices. But due to poor performance on access and linkages of care, the district scored second best in overall performance in 2016. Comparatively, the Ruth Segomotsi Mompati district had relatively high scores across sub-indices, and thus ranked highest in 2016. Thus, the balance of good performance across all sub-indices improved the overall index.

## Robustness

There was no significant difference in district ranks between index scores calculated with linearly scaled indicators (our method) and z-score standardization ($r_s$ = 0.83, 95% CI = (0.49–0.95)) (Table 2). There was also no significant difference between index scores when geometric and arithmetic aggregation techniques were used ($r_s$ = 0.95, 95% CI = (0.78–0.99)). The median absolute difference in index rankings at district level when linear and z-score standardization were compared was 2 ranks with an interquartile range (IQR) of 0–3. There was no difference in rankings (IQR = 0–1) observed at district level when indices computed with arithmetic and geometric means were compared.

All the index values across alternative indicator selections were highly correlated (Table 3).

## Validation

The C$_3$MH index had a positive correlation (r = 0.972, 95% CI = (0.63, 0.99)) with the Human Development Index (HDI) in the North West province for the period 2013–2017. The HDI measures healthy life outcomes, education and standard of living [47]. The index also increased with decreasing rates of institutional maternal mortality (iMMR) at the provincial level ($r_s$ = -0.90, 90% CI = (-1.00, -0.25)). The correlation between the index and iMMR at district level, was not statistically significant (r = -0.13, 95% CI = (-0.58, 0.39). There are no data for HDI scores at district level to allow comparisons with the COC index.

## Results of multivariate analysis

Parallel analysis in exploratory factor analysis suggested one main underlying factor for the data (Fig 5), although a three-factor model may be possible.

**Table 3. Spearman rank correlation coefficients of index values when dropping one indicator at a time to compute index.**

|  | Base case (all) | No syphilis 2&3 | No termination | No syphilis & no termination |
|---|---|---|---|---|
| **Base case (all)** | 1.00 |  |  |  |
| **No syphilis 2&3** | 0.98 | 1.00 |  |  |
| **No termination** | 0.99 | 0.97 | 1.00 |  |
| **No syphilis & no termination** | 0.98 | 0.99 | 0.98 | 1.00 |

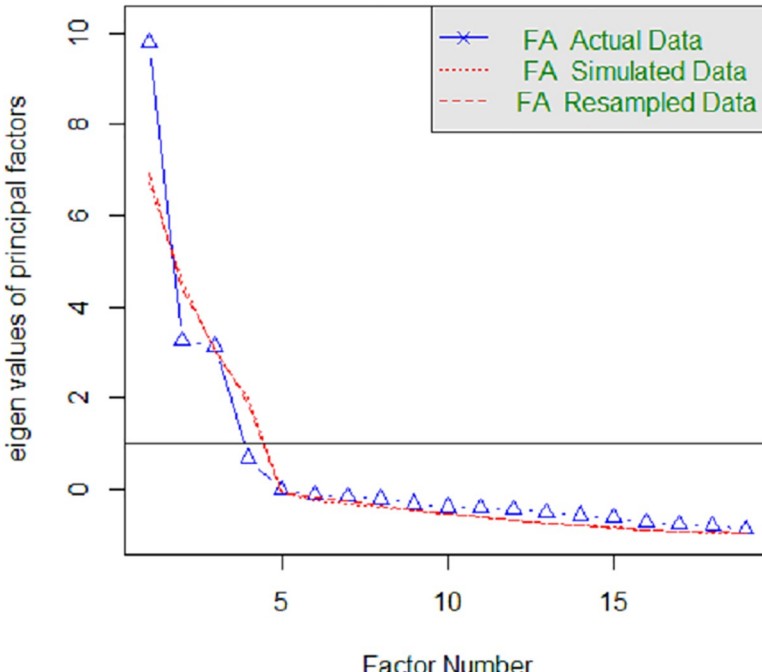

**Fig 5. Parallel analysis scree plot for indicators of the continuum of care for maternal health in North West province South Africa.**

A one factor model accounted for 0.52 proportion of variance of the data. A three-factor model accounted for cumulative variance of >0.9: the majority of factor 1 indicators related to the health system or facility based care, factor 2 contained both health system and social determinants of health, and factor 3 contained social determinants (Table 4). A two-factor model accounted for 0.72 cumulative proportion of variance of the data but did not reveal any informative conceptual groupings–all factor loadings were relatively high for one factor. The variables for linkages of care and one variable for quality of care were also not included in the results of the model due to missing data.

## Discussion

This study demonstrated the development and testing of a comprehensive and multidimensional index to assess the continuum of care for maternal health at subnational levels in SA. The $C_3MH$ index measured health and non-health sector components of service delivery for maternal health, as guided by the continuum of care framework for SA [10]. The multi-sectoral perspective of the index is increasingly important in current public health and health system performance assessment [11, 48, 49]. The index was comprehensive in that it allowed monitoring of myriad interventions, summarized through four sub- indices representing dimensions of the continuum of care. The comprehensive and multisectoral character of the index contrasts with the "silo" or vertical program approach that singles out single interventions to address maternal health outcomes [50].

Our findings suggest that the index can be used as a monitoring tool to compare subnational performance over time, and as a basis for recommendations on areas of service delivery improvement. For instance, our findings show that improvements in access and linkages to

**Table 4. Exploratory factor analysis of the items of the continuum of care service delivery framework.**

|  | Factor 1 | Factor 2 | Factor 3 |
|---|---|---|---|
| Cervical screening | **0.93** | -0.25 | 0.20 |
| Timely antenatal visit | **0.90** | -0.33 | 0.31 |
| ARTs during antenatal care | **0.95** | -0.25 | -0.21 |
| Caesarean section delivery | **0.91** | -0.08 | -0.02 |
| Delivery in facility | **0.91** | -0.12 | 0.25 |
| Couple year protection rate | **0.86** | -0.09 | -0.05 |
| HIV retest rate | **0.91** | -0.24 | 0.33 |
| Adequate food | **0.77** | 0.14 | 0.47 |
| Ante natal care coverage | -0.06 | **0.99** | -0.14 |
| Postnatal visit | 0.18 | **0.98** | 0.06 |
| Timely pregnancy termination | **-0.59** | **0.75** | 0.15 |
| Water infrastructure | -0.45 | **0.83** | -0.13 |
| Type of housing | -0.52 | **0.78** | -0.35 |
| Electricity access | -0.33 | **0.91** | -0.23 |
| Water safety compliance | -0.23 | -0.26 | **0.79** |
| Literacy | -0.37 | 0.02 | **-0.77** |
| Sanitation | 0.20 | 0.08 | **0.98** |
| Housing condition | 0.25 | 0.01 | **0.97** |
| Household dietary diversity | -0.04 | -0.45 | **0.86** |

Notes: Extraction method–ordinary least squared/minres; Rotation–varimax; Loading larger than 0.5 are in bold.

care will enhance Dr Kenneth Kaunda district's overall score and improve its ranking. On the other hand, in Dr Ruth Segomotsi Mompati district, poor performance on the social determinants of health affected the index. Thus, while the C₃MH index can be used to compare and rank districts, an analysis of sub- indices indicates areas that may proportionally affect overall performance. The relevance and utility of sub-components of composite indices for public health policy and action is important to consider as well, beyond the monitoring application of the overall index [32, 39].

Our findings also indicate that the index was robust and not much influenced by outlying scores for specific indicators. In other words, the index values the parts differently than the whole, as well as good performance over several indicators and not just a few outlying values. Further research is needed to compare the index to other comprehensive standards for the integrated care approach it seems to reflect. Alternative methods for computing the index and standardising the indicators produced comparable results. The simple geometric approach allows future integration of missing data, while maintaining the conceptual grounding of the index. Our approach also accommodates the expected refinements of indicators over time [51]; the index should undergo recurrent assessment and validation to remain useful [52]. Additionally, exploratory factor analysis indicates a distinction between at least 2 factors, with one factor covering health systems indicators and the other factor(s) social determinants of health. This reflects the multidimensional nature of the index and underlines the need to include social determinants as a dimension of continuum of care in maternal health.

Future research should also reflect emerging insights on determinants of health and wellbeing in maternal health, in order to continue to refine the framework and index. These include partner dynamics that influence a range of outcomes during pregnancy [53–57], as well as maternal mental health, gender power relations and intersectional factors such as age and (dis) ability as found in our previous research [23].

## Limitations

This was a case study of five subnational geographical areas over a five-year period. We recommend more research across other provinces/districts to allow further comparison. In other countries, the same approach using a comprehensive set of available indicators can be used to develop a contextualised version of the index. The composition of the index in this study was affected by gaps in data availability common in the SA health system [58]. We recommend health information system improvements in monitoring the quality and availability of data so that better estimates of the index can be made in the future. In addition, the lack of comparability of provincial and district level associations between the index and maternal outcomes warrants further investigation. There may be systemic issues with maternal mortality estimation in the country [59], and we also recommend use of the GHS as a source of data at district level. Other indicators could also be considered as proxies based on their shown reciprocity with maternal health indicators, such as neonatal and child health outcomes [25, 26, 60, 61].

## Conclusion

This study shows the feasibility to monitor and assess service delivery for the continuum of care for maternal health using indicators from different sectors with a composite index. The index allows monitoring of performance over time and across geographical areas. From our analyses, we concluded the index to be robust and valid, with potential to guide policy and planning to improve maternal health outcomes and service delivery from a multisectoral perspective. Future steps include engagement of health system and other relevant management stakeholders to discuss the current method and results, in order to continually refine the approach and inform any future implementation. More comprehensive monitoring of social determinants at district level and health information system strengthening can further improve and extend the use of this index. The index is amenable for testing with data from different South African and international contexts.

## Supporting information

**S1 Table. Indicators used for measurement of the continuum of care index in North West province and districts 2013–2017.**
(DOCX)

## Acknowledgments

We would like to thank the reviewers for their time and inputs into the original manuscript.

## Author Contributions

**Conceptualization:** Mamothena Carol Mothupi, Jeroen De Man, Hanani Tabana, Lucia Knight.

**Data curation:** Mamothena Carol Mothupi.

**Formal analysis:** Mamothena Carol Mothupi.

**Funding acquisition:** Mamothena Carol Mothupi.

**Investigation:** Mamothena Carol Mothupi.

**Methodology:** Mamothena Carol Mothupi, Jeroen De Man, Hanani Tabana.

**Project administration:** Mamothena Carol Mothupi.

**Resources:** Mamothena Carol Mothupi.

**Software:** Mamothena Carol Mothupi.

**Supervision:** Jeroen De Man, Hanani Tabana, Lucia Knight.

**Validation:** Mamothena Carol Mothupi, Jeroen De Man.

**Visualization:** Mamothena Carol Mothupi, Jeroen De Man.

**Writing – original draft:** Mamothena Carol Mothupi.

**Writing – review & editing:** Mamothena Carol Mothupi, Jeroen De Man, Hanani Tabana, Lucia Knight.

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
