## [Decision Letter · Decision Letter 0]

18 Mar 2021

PONE-D-20-15418

Development and testing of a composite index to monitor the continuum of maternal health service delivery at provincial and district level in South Africa

PLOS ONE

Dear Dr. Mothupi,

Thank you for submitting your manuscript to PLOS ONE. After careful consideration, we feel that it has merit but does not fully meet PLOS ONE’s publication criteria as it currently stands. Therefore, we invite you to submit a revised version of the manuscript that addresses the points raised during the review process.

Two experts in the field reviewed your manuscript. Please ensure to thoroughly address all reviewre comments.

We look forward to receiving your revised manuscript.

Kind regards,

Susan Hepp

Academic Editor

PLOS ONE

Journal Requirements:

Reviewers' comments:

Reviewer's Responses to Questions

**Comments to the Author**

1. Is the manuscript technically sound, and do the data support the conclusions?

Reviewer #1: Yes

Reviewer #2: Partly

2. Has the statistical analysis been performed appropriately and rigorously? 

Reviewer #1: Yes

Reviewer #2: Yes

3. Have the authors made all data underlying the findings in their manuscript fully available?

Reviewer #1: Yes

Reviewer #2: No

4. Is the manuscript presented in an intelligible fashion and written in standard English?

Reviewer #1: Yes

Reviewer #2: Yes

5. Review Comments to the Author

Reviewer #1: The manuscript seems scientifically sound, written in standard English language, results and findings are presented in appropriate fashion and concluded appropriately. This must be a product of years of handwork of investigators. Moreover, the study is novel and generated strong scientific evidence for the policy change and reform in maternal health in Africa and other similar countries of the world.

Reviewer #2: please see additional comments attached

Data from the DHIS is confidential and this point is made by the authors

Major issues regarding selection of indicators, lack of comment on quality of the DHIS data etc

6. PLOS authors have the option to publish the peer review history of their article (what does this mean?). If published, this will include your full peer review and any attached files.

Reviewer #1: **Yes: **Dr. Hari Prasad Kaphle, MPH, PhD, Assistant Professor (Public Health) and Coordinator (Master of Public Health Program), School of Health and Allied Sciences, Pokhara University, Kaski, Nepal. Email: harikafle07@gmail.com

Reviewer #2: No

---

## [Author Response · Author response to Decision Letter 0]

24 Apr 2021

Reviewer 1

Thank you for your time and input into our manuscript.

Reviewer #1: The manuscript seems scientifically sound, written in standard English language, results and findings are presented in appropriate fashion and concluded appropriately. This must be a product of years of handwork of investigators. Moreover, the study is novel and generated strong scientific evidence for the policy change and reform in maternal health in Africa and other similar countries of the world.

Thank you very much for your review!

Reviewer 2

Thank you for your time and input into our manuscript. Please see the changes requested in the ‘Revised Manuscript with Track Changes'. We start here with general comments then the line by line comments made below. 

This is a very important study and potentially adds to the literature and can be very useful to programme managers. It’s a brave attempt especially given issues related to data availability and quality in South Africa and in many LMICs.

Thank you, indeed data availability and quality is an issue in LMICs and the ways in which it has affected the measurement of the index and the future improvements are noted in the manuscript. 

In their review of the literature I would request the authors to pay more attention to issues related to quality of care and outcomes (these issues are reflected in my notes in the text of their manuscript).

Thank you, we have responded to the notes in the line by line comments below. 

There are areas of concern with the methodology that I wish to flag for the considerations of the authors. The author should be add a critical review of the use of linear transformation (see Greco et al, 2019 for example). In addition, the selection of indicators used (especially those from the DHIS) and their decision not to include mortality (maternal and neonatal mortality) as outcomes should be clearly stated. The latter is only noted in the section on limitations of the study. Finally, the quality of the DHIS data (in additional to imputation of missing data) should be reflected on.

Thank you, these issues were captured in the line notes made by the reviewer and we have responded accordingly to each in the line by line responses below. 

On the practical use of the index – the authors should inform the reader if the study and its approach, results and conclusions were discussed with the local managers and if any of their recommendations were adopted/implemented. I am aware that this is not usually requested by reviewers but practical application of such studies are important to improve quality of care.

Thank you, we have added in the conclusion a plan for further engagement of stakeholders (Line 354-356).

I would therefore encourage the authors to strengthen the aspects reflected above and as reflected in my notes in their manuscript.

Thank you, please see below on pages 3-8 of this letter how we have responded to and addressed the reviewers’ comments citing lines where changes have been made in the manuscript. We believe they were helpful and strengthened the manuscript. 

Major issues regarding selection of indicators, lack of comment on quality of the DHIS data etc.

Thank you we respond below to these points as they are raised in the specific lines in the manuscript and your letter.

Abstract:

Line 25: the focus was one province with the possibility of use nationwide.

Please see Line 25 and 26 for the change that reflects focus on one province with possibility of nationwide application.

Background:

Line 56: given this difference maybe add the distinction between these two measures.

Please see lines 57 to 60 added to distinguish between maternal mortality ratio and pregnancy related death ratio. 

Line 57: add administrative causes as well (patient and healthcare worker related factors)

We have modified Line 62-66 to show that these other factors contribute to deaths as well. We regarded health care worker related factors as related to the healthcare access and quality issues.

Line 63: also need to focus on regional and tertiary hospitals 

Please see Line 66-67 we have modified to include. 

Line 70: maternal nutrition is also a major contributor to good pregnancy outcomes.

Indeed, and it is featured in the framework as well, we have stated it more explicitly now in Line 78.

Line 72: The co-production of health and wellness (with in this case the pregnant woman and the partner) is surfacing as a new approach, consider adding this to the literature review?

In the paragraph we focus on explaining elements of the framework as it stands. We have discussed co-production as a new approach that could possibly inform future iterations of the framework in the Discussion (Lines 329-333). The literature on co-production of health and wellness seems to suggest partner dynamics (including abuse) that influence a range of pregnancy related attitudes, behaviours and outcomes such as pregnancy wantedness (Kroelinger & Oths, 2000), smoking cessation (Flemming et al., 2015), exclusive breastfeeding (Moraes et al., 2011), postnatal depression (Ludermir et al., 2010), alcohol use (Van der Wulp et al., 2013) among others. These were also cited by South African experts in our previous research and recommend to include them in this revision (Lines 329 - 333 in the Discussion section). 

Line 75: need to include quality of clinical care as well.

Please see Line 80 quality of clinical care added.

Line 78: add quality of care

Please see Line 84-85 quality of care added 

Line 80: quality antenatal care, improvement of clinical quality of care as well, such as ESMOE, prevention of pregnancy associated hypertension etc.

Please see Line 87 - 88 for this addition

Line 124: I would add quality of data, especially routinely collected data like the DHIS 

Please see this addition on Line 131-132

Line 133: you may wish to comment on the quality of DHIS data 

Please see Line 342 under Limitations, and Line 147 to 149 under Methodology where we include this reflection on DHIS data quality and make the limitation more explicit in the relevant section. 

Line 140: these are important indicators, please include reasons why they were not included

Please see Line 157-158 for explanation – “There were no comparable indicators of dietary behaviour and empowerment across the data sources at the district level, so only those applicable across were included.”

Line 144: with respect to poor quality data, was any data cleaning done?

The Department of Health Northwest gave us already tabulated data on specific indicators, not raw data. The only corrections made were in the rare cases where proportions were over 100%, which were revised back to 100. The DOH also reported data quality issues around data capturing rates at primary health care and hospital level in the different districts of the North West. For 2013, these figures were not there for most districts and the reported figures pointed to low data capturing rates. From January 2014 it was reported in more districts, the rates generally improved over time and by 2017 median rates were over 75%. Completeness also improved over time. With continual improvement in reporting, completeness, and data availability over time – i.e., a strengthened health information system, we believe measurement of the index will be more reliable. The authors can control for missing data through a transparent approach shared in the paper, and other elements recommendations made on strengthening the health information system. 

In line 157-158 we mention underperformance in data collection and reporting. Please see lines 158-164 for additional integration of the statements on data quality as stipulated in this comment. 

Line 158: It will be useful to add a short review of the benefits and challenges of using linear transformation, see for example Greco et al 2019. 

We thank you for this useful reference. Greco et al 2019 provide interesting guidance on weighting and aggregation. We added a more detailed justification regarding the choice of our weighting procedure (i.e. equal weights) (see lines 197-199). For aggregation, we used geometric aggregation rather than linear aggregation as Greco et al. suggest being the better choice in our case. 

The linear transformation we referred to was used for the scale transformation. 

We have modified line 180 to clear the potential confusion with linear aggregation methods and specify that it is a variable/indicator scale transformation approach, developed by the authors based on the logic of similar models with the authors’ modifications: Similar methods are referred to as linear scaling transformation (Booysen et al 2002) (doi: 10.1136/bmjqs-2018-007798), linear interpolation (Barclay et. al. 2019 – doi: 10.1136/bmjqs-2018-007798), and scale transformation (OECD 2008 10.1111/jgs.13392). We tested the approach we developed against an established method for normalization, which is z-score standardization to assure its' robustness. 

Line 200: Why were no outcome measures such as maternal mortality and neonatal mortality rates included? Surely this is the real test of performance of MH services – that pregnant women survive and deliver healthy neonates? This would have been more useful than including say cervical cancer screening coverage – the point is the authors should make a case for selection of indicators. 

Thank you. Maternal mortality and neonatal mortality rates are considered outcome indicators that do not correspond directly to process, inputs, outputs etc. in service delivery. The continuum of care framework as it is designed is concerned with comprehensive delivery of specific interventions at different levels and across the health needs of mothers. Please see Figure 1. Thus the focus of the framework is the access, utilization and quality aspects of interventions. The outcomes of maternal and neonatal mortality are indeed important, and they are used by the authors to validate the measurement of performance with the service delivery indicators. The outcome indicators thus go hand in hand with the index, where one of the validation steps prescribed by the authors looks at the correlation between the index and maternal health outcome indicators. Please see Lines 275-282 (Validation section of Results). Additionally, we make a recommendation that neonatal and child health indicators be tested in future research, please see Lines 349 - 351 of the Discussion section. 

We also include this explanation in the updated section on Selection of Indicators, please see lines 138-142.

Line 210: The more usual spelling is antiretroviral

Thank you, please see the change on Line 234

Line 213: This is why a careful explanation of indicator selection is important 

In the selection of variables/indicators section, we have added lines 136-142 to more clearly explain indicator selection. This follows along the explanation given above for non-selection of outcome indicators, as well as a point about availability. In line 136-137 we also referred to two other publications where we documented the systematic process of selecting indicators: Mothupi MC, Knight L, Tabana H. Review of health and non-health sector indicators for monitoring service provision along the continuum of care for maternal health. BMC Res Notes. 2020;13(151) and evaluating whether they were suitable: Mothupi MC, Knight L, Tabana H. Improving the validity, relevance and feasibility of the continuum of care framework for maternal health in South Africa: a thematic analysis of experts’ perspectives. Heal Res Policy Syst [Internet]. 2020 Feb 26 [cited 2020 Oct 10];18:28. Available from: https://health-policy-systems.biomedcentral.com/articles/10.1186/s12961-020-0537-8

Line 219: was this programme implemented in 2013/14?

The program was undergoing early conceptualization and testing during this time, please see the updated Line 243-244. Even though the program was not being widely implemented, we mention this data gap to explain why data is not reflected for our period of interest 2013-2017. 

Line 321: Not including outcome measures to me is the biggest limitation of the study

Outcome measures were treated as an important indicator for validation of the index – please see the methodological explanation of how outcome indicators were used in Line 217-219, and Lines 278-281 (Validation section of Results) for findings of the correlation between the index and maternal health outcome measures. The indicators aligned with the continuum of care framework used in this study reflect intervention/services and thus inputs, process and outputs and not health outcomes. Please see the added lines 138-142 on selection of indicators. 

Line 328: was the study shared with the provincial and the local management? How did they interpret the results? Was any action taken or considered by these managers on the basis of this study or will this be done should this study be published? 

The preliminary steps of the study, which are selection and assessment of indicators was done through consultations with some Department of Health and other relevant sectors’ stakeholders, which gave opportunity to get their feedback on the framework on which the index development detailed in the manuscript is based. Those findings were published elsewhere and shared with the provincial management. As follow-up and concerning this particular manuscript, we have added the engagement that still needs to happen to further refine, get interpretation, and build on the approach in Lines 358-360. The publication of the study is intended for the publication of the methodology but we acknowledge the engagement that still needs to happen to further refine the approach and get stakeholders’ further input in it. 

In their review of the literature I would request the authors to pay more attention to issues related to quality of care and outcomes (these issues are reflected in my notes in the text of their manuscript).

Thank you we have taken consideration of these and made a line by line response above.

-

Thank you again for the opportunity to revise this manuscript; and we hope that the changes made will be satisfactory and have improved it significantly. 

Yours sincerely, 

Mamothena Carol Mothupi

---

## [Editor Report · Decision Letter 1]

12 May 2021

Development and testing of a composite index to monitor the continuum of maternal health service delivery at provincial and district level in South Africa

PONE-D-20-15418R1

Dear Dr Mothupi

We’re pleased to inform you that your manuscript has been judged scientifically suitable for publication and will be formally accepted for publication once it meets all outstanding technical requirements.

Kind regards,

Yogan Pillay

Guest Editor

PLOS ONE
---

## [Editor Report · Acceptance letter]

17 May 2021

PONE-D-20-15418R1 

Development and testing of a composite index to monitor the continuum of maternal health service delivery at provincial and district level in South Africa 

Dear Dr. Mothupi:

I'm pleased to inform you that your manuscript has been deemed suitable for publication in PLOS ONE. Congratulations! Your manuscript is now with our production department. 

Kind regards, 

on behalf of

Dr. Yogan Pillay 

Guest Editor

PLOS ONE